# A Comprehensive Review of Our Understanding and Challenges of Viral Vaccines against Swine Pathogens

**DOI:** 10.3390/v16060833

**Published:** 2024-05-24

**Authors:** Aman Kamboj, Shaurya Dumka, Mumtesh Kumar Saxena, Yashpal Singh, Bani Preet Kaur, Severino Jefferson Ribeiro da Silva, Sachin Kumar

**Affiliations:** 1College of Veterinary and Animal Sciences, G. B. Pant University of Agriculture and Technology, Pantnagar 263145, Uttarakhand, India; aman.kamboj.biotech@gmail.com (A.K.); mumteshsaxena@gmail.com (M.K.S.); yash.singh034@gmail.com (Y.S.); 2Department of Biosciences and Bioengineering, Indian Institute of Technology, Guwahati 781039, Assam, India; shauryadumka98@gmail.com (S.D.); b.preet@alumni.iitg.ac.in (B.P.K.); 3Leslie Dan Faculty of Pharmacy, University of Toronto, 144 College Street, Toronto, ON M5S 3M2, Canada; jefferson.silva@utoronto.ca

**Keywords:** pigs, viral diseases, vaccines, new generation vaccines, adjuvants, vaccine delivery

## Abstract

Pig farming has become a strategically significant and economically important industry across the globe. It is also a potentially vulnerable sector due to challenges posed by transboundary diseases in which viral infections are at the forefront. Among the porcine viral diseases, African swine fever, classical swine fever, foot and mouth disease, porcine reproductive and respiratory syndrome, pseudorabies, swine influenza, and transmissible gastroenteritis are some of the diseases that cause substantial economic losses in the pig industry. It is a well-established fact that vaccination is undoubtedly the most effective strategy to control viral infections in animals. From the period of Jenner and Pasteur to the recent new-generation technology era, the development of vaccines has contributed significantly to reducing the burden of viral infections on animals and humans. Inactivated and modified live viral vaccines provide partial protection against key pathogens. However, there is a need to improve these vaccines to address emerging infections more comprehensively and ensure their safety. The recent reports on new-generation vaccines against swine viruses like DNA, viral-vector-based replicon, chimeric, peptide, plant-made, virus-like particle, and nanoparticle-based vaccines are very encouraging. The current review gathers comprehensive information on the available vaccines and the future perspectives on porcine viral vaccines.

## 1. Introduction

According to the United Nations Population Division, 2019, the human population is expected to rise to around 8.5 billion in 2030, 9.7 billion in 2050, and 10.4 billion in 2100 [1]. The reduction in agricultural land and global climate change make it more challenging to ensure food security for the human population, which is increasing at an alarming rate. The livestock sector can play a significant role in ensuring food security as animal-based foods provide complete nutrition with high-quality major dietary proteins. It is evident that parallel to the global population growth, the total meal production and global meat consumption also increased. With the increased demand for animal proteins, pork contributed as one of the major sources of proteins, and pig farming emerged as a significant livestock farming practice in several developed and developing countries. As per the latest data, the total global pig population was approximately 780 million, which increased from 750 million heads in 2021 [2]. China is the home of the largest number of pigs, i.e., around 450 million heads, followed by the European Union and the United States. With the increasing population, the consumption and trade of pork also increased in the global scenario; however, the swine industry suffers from many challenges worldwide.

Infectious diseases constantly challenge pig farming, leading to annual economic losses worldwide. Viral infections are at the forefront of these diseases, causing a huge economic burden due to mortality and production losses. In the last few years, several viruses have emerged and reemerged, some of which cause severe clinical disease, whereas others have a less or negligible impact on pig farming. The pig industry has witnessed a massive transformation in the last few years after the outbreak of viral diseases of broad economic concern. One such example is the occurrence of African swine fever (ASF) in China, as it contributes to half of the global pig population. Since the first report of ASF in China in the year 2018, more than 1 million pigs have been culled up to 2020, with a recorded drop in pork production by 55% [3]. The count may have become more alarming due to repeated outbreaks reported in China since then. A comprehensive analysis of publication trends from 1996 to 2016 highlighted ASF, classical swine fever (CSF), foot and mouth disease (FMD), porcine circovirus infection (PCV), porcine reproductive and respiratory syndrome (PRRS), pseudorabies, swine influenza, and transmissible gastroenteritis (TGE) as crucial porcine viral diseases [4].

Vaccination is considered the most effective and economical way to prevent viral infections. It provides active artificial immunity that stimulates either humoral and/or cellular immune responses. From the 18–19th centuries to the present day, the field of vaccine technology has witnessed enormous growth and development [5]. According to the development timeline, vaccines are classified as first-, second-, and third-generation. The first-generation vaccines constitute the whole organism, including the killed (inactivated) and live attenuated pathogens. These vaccines are crucial in controlling, preventing, and eradicating viral diseases in humans and animals, enhancing livestock productivity, and promoting food security by reducing morbidity and mortality [6,7]. Some important examples include vaccines against smallpox, polio, and rinderpest. However, these first-generation vaccines have dominated the field of immunization for more than a century but still suffer some disadvantages that need to be worked upon. The era of second-generation vaccines started with the subunit components of the pathogen, like purified antigenic proteins, recombinant antigens, and synthetic proteins. The second-generation vaccine technology does not rely on whole viruses or organisms but utilizes the protein expression and purification platforms to produce overexpressed, purified antigens. Like first-generation vaccines, second-generation vaccines have their advantages and disadvantages, which directed the newer vaccine development technologies into the timeline of exploration [5,6,7]. The advent of genetic engineering tools and an improved understanding of antigen structure paved the way for developing second- and third-generation vaccines [8]. The third generation of vaccines presents platforms like DNA, mRNA, viral vector-based, and chimeric antigens.

The development in the field of molecular biology enables us to manipulate genetic material, which can be easily used to devise alternate strategies for the production of new-generation vaccines [9,10,11,12]. Apart from vaccine technologies, much work has also been carried out in its delivery and administration [6,12,13]. The history of veterinary vaccines encompasses more than three centuries, from the initial attempts of variolation in the 1700s to the development of the most recent COVID-19 vaccine for animal use. The products and milestones in animal vaccine development have been shown in Figure 1.

## 2. Types of Vaccines

Vaccines can be categorized based on their constituents or the technologies used for their development. Several types of vaccines are commercially available or in development for use against porcine viral diseases (Figure 2).

## 3. Whole-Organism Vaccines

Historically, conventional vaccines were developed based on an empirical trial and error approach, which simulates the natural infection to induce immunity in the host. It relies on the conventional “isolate, kill, or inactivate and inject” methodology, which forms the basis of traditional immunization [14,15]. This approach uses the whole organism rather than the antigenic entity. This approach is also used for those pathogens that can be easily cultured in vitro. These vaccines have been crucial in improving animal and public health since their inception; most licensed veterinary vaccines still fall under this category [6]. Whole-organism vaccines have been successfully tested and used against most of the economically significant swine viral diseases, including swine influenza, PRRS, pseudorabies, porcine endemic diarrhea, PCV infection, etc. [16].

### 3.1. Conventional Killed/Inactivated Vaccines

Conventional killed/inactivated vaccines are some of the oldest developed vaccines, consisting of an entire pathogen that has been killed or inactivated by physical or chemical means so that it is no longer pathogenic to the host. The safest among most types, these vaccines gained importance and emerged for use in pigs after the Second World War. The safety of these vaccines is attributed to their inability to revert to a virulent virus after inactivation. The virus is generally grown in cell culture and inactivated using different methods [16,17]. The commonly used physical method of inactivation is thermal treatment, which uses heat and radiation. On the other hand, the chemical inactivation method generally uses formaldehyde or beta-propiolactone [18]. The inactivation causes the denaturation of viral proteins or damage to their genome. Despite their better safety and stability, killed vaccines suffer several limitations such as short-term immunity, poor cellular and mucosal immunity, the need for boosters, the requirement for an adjuvant, and high production cost [19].

Although some limitations are associated with inactivated vaccines, they are essential to the control program for swine viral diseases. Multiple swine viruses have been a matter of grave concern globally, including the emergence of novel genotypes and serotypes [20]. Influenza A virus (IAV) is an important swine pathogen mainly controlled by the inactivated vaccine [21]. Control of swine influenza with the commercial whole inactivated virus (WIV) vaccine has proven effective against genetically similar or homologous viruses. However, it lacks heterovariant and heterosubtypic protection against swine influenza viruses [22]. Maternally derived antibodies also affect the efficacy of WIV vaccines. Another complication reported with the use of WIV against swine influenza is the occurrence of vaccine-associated respiratory diseases in vaccinated pigs [22]. Vaccine-associated enhanced diseases (VAED) are altered forms of clinical infections observed in individuals who have previously been vaccinated against a specific pathogen and subsequently exposed to the wild-type version of that pathogen. Classic instances of VAED include atypical measles and enhanced respiratory syncytial virus (RSV) infections following the administration of inactivated vaccines for these diseases [23]. Due to these WIV vaccine-associated limitations, alternate strategies with attenuated and new-generation vaccines have been tried against the swine influenza virus.

### 3.2. Live/Attenuated Vaccines

Live/attenuated vaccines are the most widely used class of vaccine types, typically based on the traditional concept of weakening the pathogen’s virulence and administered to the host as a vaccine. During the attenuation process, the virus is passaged until the host stops showing clinical signs of disease or the cytopathic effects no longer appear in the cell culture [24]. The field of vaccinology witnessed immense developments in the past few decades with the emergence of various cutting-edge technologies. However, live attenuated vaccines are still the first choice due to their numerous advantages over the other types [6,7]. These vaccines are best suited for single doses with multiple routes of administration, such as intramuscular, intradermal, intranasal, or oral. The live attenuated vaccine mimics the natural infection, eliciting a strong humoral and cell-mediated immune response with occasional mild clinical signs of the disease. It does not require any adjuvant and provides relatively long-term immunity with low production costs. However, safety and storage are two significant concerns. There is always a chance of reversion of the attenuated virus to a virulent form. The live vaccine must be stored at −80 °C or in lyophilized form and cannot be stored once opened [16,25]. Some major pitfalls associated with safety include the use of attenuation methods that are random and non-specific, and the extent of attenuation is not regulated [25].

Advances in synthetic biology allow researchers to overcome these two bottlenecks associated with the development of live attenuated vaccines. Synthetic attenuated virus engineering (SAVE) relies on decreased gene expression by introducing deoptimized codon pairs using a computer algorithm. Due to codon deoptimization, such attenuated viruses can produce proteins with wild-type amino acids but with low efficiency, which enables them to overcome host defense without sufficient replication. These synthetically attenuated viruses are excellent vaccine candidates as they mount a robust immune response with long lasting protective immunity. This approach has been successfully applied for the rapid attenuation of the PRRS virus where the major envelope GP5 gene was codon-pair deoptimized, aided by a computer algorithm. Finally, the codon-deoptimized vaccine named SAVE5 was rescued in vitro and shown to have a lower replication rate and reduced expression of GP5 protein. In vivo studies revealed that SAVE5-infected pigs showed a lower level of viremia up to 14 days post-infection with reduced gross and histological lung lesions compared to the wild-type virus [26].

Classical live attenuated viral vaccines have paved the way for developing new-generation vaccines, but they are still in use for viral diseases like PRRSV [27]. Safety concerns, however, are not common. The outbreak of PRRS in Danish Pigs in 2020 due to the recombination of two vaccine strains raised safety concerns [28,29,30]. Current commercially available PRRSV live attenuated vaccines include Ingelvac^®^ PRRS MLV and Ingelvac^®^ PRRS ATP (BIVI), ReprosCyc^®^ PRRS EU (BIVI), Porcilis^®^ PRRS (MSD), Fostera^®^ PRRS (Zoetis), Suvaxyn^®^ PRRS MLV (Zoetis), Prime Pac^®^ PRRS RR (Merck), Prevacent^®^ PRRS (Elanco), UNISTRAIN^®^ PRRS (HIPRA), and AMERVAC^®^ PRRS (HIPRA) [16,31] (Table 1). An NS1 truncated bivalent (H1N1 and H3N2) live attenuated influenza virus (LAIV) is commercialized for use in the USA as Ingelvac Provenza™ (Boehringer Ingelheim, St. Joseph, MO, USA) in piglets at one day of age through the intranasal route. The experimental pieces of evidence of this vaccine highlight reduced IAV shedding in pigs challenged 12 days post-vaccination [32].

Live attenuated vaccines (LAVs) are also used to control ASF and CSF [33]. Recent developments have sparked optimism regarding the potential for an effective and safe vaccine. One notable candidate is “ASFV-G-∆MGF”, a live attenuated vaccine that has shown promise in preclinical studies [34]. It was observed that the pigs infected with the African swine fever virus (ASFV) develop protective immunity against new infections after survival. This indicates that it might be possible to develop an effective vaccine against ASFV [35]. However, the attempts made to develop the inactivated, recombinant protein and DNA-based vaccine failed even when combined with the specific adjuvants [36,37,38]. Therefore, LAV emerges as a crucial strategy to develop an effective vaccine against ASFV. However, until 2021, no commercial vaccine was available for ASF. According to a report, the first vaccine against ASF NAVET-ASFVAC of the National Veterinary Medicine Joint Stock Company (NAVETCO) was recently commercialized in Vietnam [39]. Despite limited reports on commercial ASF live vaccines, there are several reports of the development of ASFV LAVs with a variable degree of protection from 0 to 100% using vaccine candidates derived from naturally attenuated (NH/P68; OURT/88/3) and virulent (Georgia07; Benin 97/1; Ba71) strains [40,41,42,43,44].

In many areas worldwide, LAVs were widely used against CSF. They paved the way to its eradication but suffered a major drawback due to their inability to differentiate the infected from vaccinated animals (DIVA). Among LAVs against CSF, C-Strain is the most widely used and is effective against all the genotypes of CSFV [45,46]. The attenuated vaccine strains of CSFV are generally produced by serial passages either in rabbit (lapinized) or in cell culture. Other than lapinized C-strain, French cell culture adapted thiverval, Lapinized Philippines Coronel (LPC), the low-temperature adapted Japanese guinea pig exaltation-negative (GPE-) strain, Mexican PAV strain, and LOM strain are also used as LAV candidates [47,48]. The research thrust is diverted more toward the production of cell culture-based LAVs due to the ethical and scalability issues. However, the high cost of production and administration challenges in resource-poor areas contribute to the continuous circulation of the virus in the vaccinated population. It is, therefore, seen that prolonged suboptimal vaccination leads to changes in the pathogenicity and antigenicity of CSFV strains, which could lead to escape variants [49,50,51]. In addition to C-strain for vaccinating domestic pigs, there are reports of its use as an oral vaccine in wild boars and domestic pigs in rural settings [52,53]. Unlike injectable vaccines, oral preparations are more effective and easier to administer. The combination of mucosal immune responses, enhanced immune activation, broader immune protection, and suitability for mass vaccination makes oral vaccines highly effective. Tremendous improvements have occurred in formulating oral vaccines including the baits by absorption of C-strain onto bread followed by subsequent lyophilization [54]. It is stable for 18 months at 4 °C, contrary to the commercial oral vaccine containing liquid C-strain, which requires −20 °C for storage. It is critical to use an oral vaccine and monitor the antigen–antibody response among the population to control CSFV in wild boars [55].

### 3.3. Live Attenuated DIVA Vaccines

Conventional live attenuated and inactivated vaccines are effective in providing protection but lack DIVA specificity. The vaccines that allow easy and precise differentiation between infected and vaccinated animals serologically are known as marker vaccines. Among these, recombinant chimeric viruses and gene-deleted marker vaccines are important [56,57,58]. Chimeric viruses also present a powerful platform where the backbone of one virus is used, along with the genes of another virus [59]. Among chimeric pestivirus candidates, the most promising ones are infectious cDNA clones of CSFV or bovine viral diarrhea virus (BVDV) [60]. Another marker vaccine is the CP_E2alf, which was licensed as the first live marker vaccine against CSF with the name “Suvaxyn^®^ CSF Marker” by Zoetis and approved by the EU for emergency vaccination within restricted control [61]. Another chimeric pestivirus, CP7_E2gif, has also been reported over time, which contains a BVDV backbone without any gene of CSFV; instead, BVDV envelope protein E2 has been replaced by its homolog from border disease virus (BDV) [62,63]. The PRV HD/c vaccine is a gene-deleted marker vaccine designed to control Aujeszky’s disease by removing specific gE/TK genes [64]. Notably, the absence of these genes allows for differentiation between natural infection and vaccination [64].

### 3.4. Viral Vectored Vaccine

The recombinant virus is as efficient as the live virus and retains the other advantages of live vaccines with better safety [65,66,67]. The working principle of the viral vector vaccine is given in Figure 3. SV40 was the first viral vector created in the year 1972 for the expression of a foreign gene [68]. Since then, various other viruses have been explored and engineered for use as antigen-delivery vectors. Adeno-, herpes-, pox, and paramyxo-viruses are viral vectors commonly used in veterinary medicine, and among these, adenoviral vectors are an ideal candidate for successful vaccine delivery [69,70,71]. They offer several advantages, such as genetic safety, low pathogenicity, and non-integration into the host genome [71,72,73]. Poxvirus, which includes vaccinia, canary, and fowlpox, has also been explored successfully for carrying exogenous antigens to the hosts [74]. The live mammalian viral vectors can be classified into replicating and replication-deficient viral vectors. The replicating viral vectors can easily replicate in the infected cells, whereas the replication-deficient viruses lack the functions essential for replication and virion assembly. PRRSV has also been reported to be used as a live viral vaccine expressing antigenic proteins of IAV and PCV2, in the form of a multicomponent viral vector vaccine. It has shown a significant reduction in lung and lymphoid lesions, reducing the acute respiratory signs following the challenge with PRRSV, PCV2, and IAV [17,75].

The Newcastle disease vaccine (NDV) expressing CSFV E2 and Erns proteins proved to be an effective alternative for cost-effective vaccine production [17,76,77]. It can also be used via the intranasal route, delivering viral proteins at the primary virus entry and replication site. CSFV E2 protein has also been produced using recombinant baculovirus and swine pox virus [78,79].

## 4. Subunit Vaccines

With the advancement of immunology, a better understanding of antigen structure and immunogenic moiety has been made possible. The viral surface proteins and peptides are the leading vaccine candidates used as subunit vaccines against viral pathogens [80]. Moreover, polysaccharides and conjugated subunit vaccines can also be effective against bacterial pathogens [81]. The immunogenic proteins may be isolated from viruses or expressed using recombinant DNA technology. The subunit vaccines have several advantages over other vaccines; the foremost is its safety profile. It has no live component, so it cannot potentially revert to the infectious or virulent form. The immune response is only produced against the small immunogenic part used as an immunogen. Also, it is a well-established technology suitable for vaccinating immunocompromised hosts with utmost safety including chronic and pregnant subjects [17,80,81]. It has no side effects at the site of injection and is relatively stable and easy to transport.

Further, subunit vaccines provide a much more targeted or precise immune response against a specific immunogenic determinant. Its production is streamlined with the least variations among the processing lots [17]. Despite possessing numerous advantages, subunit vaccines still involve challenges like poor immunogenicity compared to attenuated vaccines, leading to the requirement of adjuvants and boosters to provide long-term immunity [6,16,17]. In the present scenario where eradication of an infectious disease necessarily needs a vaccine capable of allowing DIVA, the subunit vaccine has the upper hand on whole-organism vaccines as it provides the added advantage of DIVA [82]. Therefore, extensive work has been carried out on recognizing protein candidates for subunit vaccines. As the development of subunit vaccines mainly relies on the knowledge of microbial components against which an immune response is expected, the prediction of antigenic protein or peptide is highly desirable. It can be quickly performed based on the serological responses of the recovered animals. Recent bioinformatics tools are allowing us to predict antigenic determinants in silico (Table 2).

### 4.1. Purified Antigens

Purified antigens are the split products of pathogens, where specific components are used as a vaccine. They are generally divided into four main categories: protein-based antigens, polysaccharides, conjugates, and toxoids. These antigens are used for bacterial pathogens and have been less extensively explored for viruses [83]. However, for porcine viral diseases, negligible reports are available for commercial vaccines based on purified viral antigens.

### 4.2. Recombinant Proteins

With the advancement of protein expression techniques and purification systems, it has become convenient and cost-effective to express viral antigenic proteins. Viral antigens are commonly expressed using both prokaryotic and eukaryotic expression systems.

Prokaryotic Expression System: *E. coli* and *B. subtilis* are the most common prokaryotes for viral antigen expression that offer advantages like fast growth, easy scale-up, high yield, and low cost. A lack of post-transcriptional and post-translational modifications and patterns of codon usage are some of the challenges [84,85]. However, engineered strains of bacteria can overcome these disadvantages and induce specific modifications required to produce the desired protein of interest. The heterologous expression of the recombinant capsid protein of porcine circovirus 2 (PCV2) in *E. coli* and their use as subunit vaccines have been well documented [86,87]. The highly antigenic pK205R protein of the ASFV is successfully cloned and expressed in *E. coli*. A chimeric protein containing short amino acid sequences from PRRSV glycoprotein 3 (GP3), glycoprotein 4 (GP4), glycoprotein 5 (GP5), and M (matrix protein) expressed in *E. coli* showed a robust immune response in mice and piglets [88].

Eukaryotic expression system: Yeast, mammalian, insect, and plant cells are often used as an expression system. The eukaryotic expressed proteins are post-translational modifications, high scalability, and a similar codon usage pattern as occurs in the natural viral replication cycle [16].

*S. cerevisiae*, *S. pombe*, and *P. pastoris* are commonly used yeasts for producing viral protein. These vector systems show a fast growth rate, low production cost, high protein yield, and better protein modification and folding. Although the yeast system overcomes all the disadvantages encountered in the prokaryotic expression system, it is not commonly exploited for swine vaccines. It has been reported that a whole yeast vaccine expressing the S1 protein of porcine epidemic diarrhea virus (PEDV) induced a high IgA response in pigs administered with the antigen orally [89].

The mammalian expression system is the most suitable system when the production of complex molecules is required, along with precise glycosylation. However, the system suffers some disadvantages, including low-to-medium scalability, high production costs, endogenous virus contamination, and a slow multiplication rate [90]. Human embryonic kidney 293 (HEK-293) and Chinese hamster ovary (CHO) are two widely used cells for the expression of viral proteins [90]. However, like yeast expression systems, the mammalian cells have only been explored to a limited extent for porcine vaccinology. Studies on the expression of the PEDV S1 protein using three eukaryotic expression systems, including yeast, insect, and mammalian cells, showed the highest yield of glycosylated protein in the case of HEK-293 cells, and vaccinated sows and piglets showed high titers of neutralizing IgG and IgA [91]. The primary immunogenic protein E2 of CSFV has also been expressed successfully in HEK-293 and CHO cells [92,93].

Baculoviruses comprise a diverse range of DNA viruses reported to infect more than 600 species of insects but are non-pathogenic to mammals and are excellent vectors for the expression of mammalian or viral proteins [94]. *Autographa californica multiple nucleopolyhedroviruses* (AcMNPV) containing a 134 kb circular double-stranded DNA genome is the best-characterized baculovirus used as a protein expression vector in insect cells. The baculovirus vectors are safe as they neither replicate inside the transduced cells nor integrate into the host chromosomes [95]. The baculoviral expression system is widely used to produce several porcine viral subunit vaccines, some of which are experimental and commercially available. Commercial baculovirus-expression-system-based subunit vaccines are available against PCV2, porcine parvovirus (PPV), and CSFV and are listed in Table 3. The baculovirus vector has been successfully used on an experimental basis for the expression of immunogenic proteins from pseudorabies virus (PRV) (glycoprotein D), CSFV, and hepatitis E virus [96,97,98].

*Plant-derived vaccines:* Plants have been used as the host for the mass production of recombinant protein and other therapeutics [99,100]. Approval from the USDA for the production of the first veterinary vaccine in plants against Newcastle disease virus marked a significant breakthrough. It opened a new research area for the large-scale production of veterinary vaccines in plants [101]. Due to the low cost of production of veterinary vaccines, plants represent attractive biofactories that offer multiple advantages over the existing expression platforms including easy implementation, better safety, scalability, and ability to induce post-translational modifications [102].

The CSFV-E2 protein has been successfully produced in *Arabidopsis thaliana* and *Nicotiana benthamiana*, demonstrating effective immunogenicity [103,104]. Dimeric E2, produced in *N. benthamiana*, was found to provide adequate protection against CSFV when administered in a single dose with an adjuvant [104,105]. A subunit vaccine candidate against PEDV in transplastomic tobacco plants has been used for pigs [106]. Additionally, the ORF-5 gene of PRRSV has been made by transforming embryonic cells of the banana plant using Agrobacterium-mediated transformation and has shown promising results [107]. The reports are also available for the PRRSV expression in tobacco and potato plants [108,109].

*Transgenic animal system:* Animal bioreactors present another promising platform to produce veterinary subunit vaccines [110]. Vaccines could be made to be excreted in milk and eggs using advanced molecular biology tools. Reports of porcine viral subunit vaccine production in transgenic animals are limited. A highly protective E2-CSFV vaccine candidate produced in the mammary gland of adenoviral transduced goats has been shown to give long-lasting protection in pigs [111,112].

### 4.3. Synthetic Peptides

Understanding antigen and epitope structures and the advent of peptide synthesis techniques made it possible to develop peptidomimetics [113]. Synthetic peptides offer some unique advantages over the other vaccine platforms. They could mimic B- and T-cell epitopes from the infectious agent. The reports of synthetic peptide vaccines for swine viral diseases are minimal except for FMDV due to poor immunogenicity and partial protection associated with the linear peptides [114]. Multimerization of peptides opened up a new generation of synthetic peptide vaccines, where dendrimers are at the forefront. These multimeric peptide structures confer better immune responses and protection than linear peptides. Successful reports are available for peptide dendrimers containing multiple copies of B- and T-cell epitopes against FMDV or CSFV [114,115,116].

## 5. Nucleic-Acid-Based Vaccines

Nucleic acid or genetic or gene-based vaccines comprise nucleic acid (DNA or RNA), which is injected directly and taken up by cells to produce that protein, which ultimately leads to the generation of a humoral and cell-mediated immune response.

### 5.1. DNA Vaccines

DNA vaccines use a plasmid that carries genes encoding antigenic proteins of the pathogen. The naked plasmid is injected into the recipient’s muscles so that the muscle cells take up the construct to form an antigenic protein [117]. The role of dendritic cells (DCs) is crucial; they present the antigens to MHC (Major Histocompability Complex) class I and II products, which in turn present them to immune cells, leading to cytokine production and reaction to CpG oligodeoxynucleotides [118]. Therefore, DCs act as natural adjuvants, especially in the case of genetic immunization, making it a promising vaccination strategy.

DNA vaccines are also developed and experimentally evaluated for several swine viral diseases. An epitope-driven plasmid DNA (pDNA) vaccine against IAV is compared for its efficiency after intradermal administration with a commercial inactivated whole virus vaccine when administered intramuscularly and found to confer a better CMI (Cell Mediated Immmune) response than the traditional vaccine [119]. DNA vaccines encoding p30 and p54 genes of immunogenic proteins of ASFV showed substantial antibody titer in mice but failed to do so in pigs. However, it was found to be improved exponentially by adding the extracellular domain of ASFV hemagglutinin (sHA) to the vaccine-encoded antigens [120]. A DNA vaccine encoding the E2 protein gene was used against CSFV and shown to confer total protection following virulent challenge [121]. Recently, a novel bivalent vaccine containing the recombinant alphaviral plasmid encoding the E2 and Erns proteins of CSFV and Rep as well as the Cap of PCV2 showed promising results in mice [122]. The mosaic vaccines have been developed using the DNA vaccine strategy for viruses with different strains with limited cross-protection. DNA vaccine constructed with ORF5 PRRSV mosaic sequences induces an effective cellular immune response against heterologous challenge when administered with cationic liposomes [123].

### 5.2. mRNA Vaccines

Vaccination using mRNA is one of the most advanced, safe, rapidly scalable, and effective vaccine strategies, and it is safe and effective [124,125]. It uses the host cell translation machinery to produce antigenic proteins in vivo and mimic the natural infection [126]. Preliminary research data showed that when two 80 µg rabies vaccine dosages were administered 21 days apart, high neutralizing antibody titers and antigen-specific cytotoxic T cells and helper T cell responses were observed in mice and pigs [127]. mRNA vaccines based on antigens expressed by a single GP5 mRNA or fused GP2-GP5-M-mRNA of PRRSV were designed, and both vaccinations dramatically increased the cellular and humoral immune responses. When delivered at large dosages, the GP5-mRNA vaccine produced an immune response comparable to that of commercially available vaccines. However, a significant shortcoming was that it only featured mice model experiments and did not include immunization or protection against pigs [128]. In addition, mRNA vaccine candidates against PEDV have been created, and when tested in mice and pregnant pigs, they evoked immense humoral and cellular immune responses, as well as high neutralizing antibody levels. In pregnant sows, mRNA PED vaccination based on the receptor binding region (RBD) heterodimer generated neutralizing antibody levels comparable to those of the available inactivated vaccines [129]. Another PEDV vaccine that targets the spike (S) protein of PEDV induces a strong antibody response and antigen-specific T-cell responses in immunized piglets [130]. Furthermore, an mRNA-vectored Nipah virus (NiV) vaccine expressing soluble G glycoprotein candidates in pigs was developed using lipid nanoparticles. After booster immunization, pig serum had high antigen-binding and virus-neutralizing antibodies. Antibodies can also prevent glycoprotein-mediated cell–cell fusion. Specific T cell cytokine responses were also detectable following booster immunization, with evidence of both CD4 and CD8 T cell induction [131].

## 6. Emerging New-Generation Vaccine Technologies

Vaccines, such as inactivated, live-attenuated, and subunit vaccines, have been used for several decades, and all of these vaccines have their own merits and demerits [16]. New approaches are being adopted to create long-term effects in vaccines to improve the immunity and long-term protection of animals. Some new-generation vaccine technologies include virus-like particles (VLPs), dendritic cell-based vaccines, and mucosal vaccines [132,133].

### 6.1. Virosomes and Virus-like Particles (VLPs)

Virosomes constitute monolayered or bilayered phospholipid membranes with viral glycoproteins. They also act as a vehicle to deliver drugs or vaccines, as they are capable of fusion with target cells [134,135]. VLPs, on the other hand, are protein structures with multiple subunits identical to native virus particles. VLPs, owing to their nanometric size, are easily detected by antigen-presenting cells that further activate T-cells [136]. This has been used to develop vaccines against TGEV and the PRRSV [137,138]. A pseudo-viral particle expressing recombinant M and E proteins of TGEV was generated to induce a detectable interferon (IFN) response [137]. In the case of PRRSV, virosomes are considered promising and have also been reported to target plasmacytoid dendritic cells (pDCs) to induce a T-cell response [138]. While the use of virosomes as a porcine vaccine has seen little development, VLPs, on the other hand, have shown multiple successes. PED VLPs expressing M and N proteins exhibit a broad spectrum of cellular immunogenic responses [139]. The ELPylated cap protein of PCV2 generates enhanced immunity by increasing cytokine and antibody production [140]. A recombinant viral approach for forming VLPs in host cells was performed for PRV and PRRSV, where rPRV-NC56 is a pseudorabies virus with genes from the PRV variant strain XJ and NADC30-like PRRSV strain CHSCDJY-2019. It was capable of indefinitely expressing the GP5 and M proteins. The introduction of the self-cleaving 2A peptide allowed GP5 and M proteins to be expressed independently, form PRRSV VLPs, and induce cellular and humoral responses [141]. Further advancements in VLPs targeting multiple pathogens gave promising approaches for viruses such as PCV 2 and FMDV [142,143].

### 6.2. Mucosal Vaccines

Mucosal vaccines include administration through nasal and oral routes and through the eyes [144]. Target viruses for mucosal vaccines include PRRSV, CSFV, FMDV, PEDV, and PCV2, among many others [93,145,146,147,148]. Mucosal vaccines have high potential for the secretory antibody response towards viruses, as most of them cross mucus lining passages to infect various organs. Mucosal vaccines combined with appropriate adjuvants (*M. tuberculosis* WCL, *Cholera toxin B* subunit, and OK-432) have been shown to be successful in the control of viruses like influenza virus, PRRSV, poliovirus, rotavirus, parainfluenza-3 virus, and respiratory syncytial virus [93]. Live attenuated and inactivated vaccines can also be administered orally or intranasally for PEDC or TGEV [148].

### 6.3. Dendritic Cell Vaccine

Subunit vaccines carrying specific antigens from a virus can be fused with DC-targeted peptides to improve T-cell-mediated immune responses in animals [149]. Apart from peptides, vaccine adjuvants such as α-D-glucan and poly (I:C), which activate DCs, can cause enhanced immunity in animals along with the viral antigen, as shown in the case of the swine influenza virus [150]. PEDV vaccines are available as live attenuated or inactivated viruses and protein-based viral recombinant mucosal vaccines that produce substantial stimulation to the DCs to improve cytokine production [151,152,153,154,155]. ASFV might not be a well-studied pathogen concerning DC stimulation as it bypasses the APCs and actively replicates. There is potential for the development of DC-based vaccines for ASFV using better adjuvants or peptides along with classical vaccines [156]. Lastly, pig antigen-specific CD4 T cell immunity is elicited by porcine DC-SIGN in dendritic cells in PRRSV and increases interleukin production [157,158].

### 6.4. Multivalent and Polyvalent Vaccines

While most vaccines target only one virus at a time, multivalent vaccines use two or more viruses. The most common pathogens often targeted together include PCV2 with other viral and non-viral pathogens [159,160,161,162,163]. VLPs produced by the Cap protein of PCV2 are remarkably efficient against its infection. They can combat PCV2 and other pathogens by imbuing VLPs with immunomodulating peptides or proteins from other viruses, such as IAV, FMDV, or PRRSV, involving spycatcher/spytag technology [142]. Sometimes, the bivalent nature of vaccines supports more than one protein of the same virus, as in the influenza virus, against hemagglutinin and neuraminidase to improve efficacy [164]. Another example of recombinant virus vaccines targeting more than one virus is rPEDV-PoRV-VP7 virus containing the VP7 protein of porcine rotavirus [165]. Lastly, multi-epitope vaccines using poly (I:C)-containing epitopes of the same virus are effective for conferring cross-protection in the case of FMDV [166].

## 7. Reverse Genetics and Personalized Vaccines

Reverse genetics (RG) technology exploits the idea of working our way down from genomics to proteomics. Here, genomic sequences that contain information on antigenic proteins are used to develop a vaccine [167,168]. PRRSV and porcine influenza viruses are exploited for reverse genetics and have been used to generate modified viral antigens [169,170,171]. The PRRSV reverse genetics system is a foundation for the reasonable and logical creation of an innovative PRRSV vaccine. PRRSV immunological variants were created using random sequence rearrangement. RG is an important technique for producing genetically modified RNA and DNA viruses from cDNA copies. This technique is widely used in influenza virus studies to understand better various aspects of influenza multiplication, pathogenicity, propagation, and vaccine development [171]. RG is applied for PEDV, a pestivirus, among others, and can be a step forward in generating better vaccinations [172,173]. Assessment of an individual’s genetic background, sex, and other variables that could influence vaccine antigenicity, potency, and safety is the foundation of personalized vaccinology. A personalized vaccinology approach suggests developing specific vaccines based on factors related to overcoming the potential for poor immunogenicity and adverse events. Influenza vaccines are an excellent example [174,175]. While such an approach has not been used to develop any animal vaccine, the field is promising and can be translated into porcine vaccines, and subsequently for various other viral pathogens.

## 8. Vaccine Administration Routes

Vaccine administration is one of the most critical concerns in vaccinating domestic and wild animals. A variety of traditional and alternative administration routes are employed for vaccinating pigs. A promising vaccine administration route facilitates fast immunization, is less labor-intensive and non-invasive, and causes minimum stress to the animal [24]. In domestic animals, intramuscular (IM) and subcutaneous (SC) routes are the most common routes of vaccine administration, while in pigs, IM route is the preferred route [16,24,176]. Recently, the traditional IM route of administration has been replaced by intradermal, intranasal, or oral routes [24,177,178]. Intradermal routes for DNA vaccines have shown promising results in the porcine industry. Needleless devices are available in the market, and intradermal administration by these devices has reduced pain and fear in sows [16,177,178]. Except for the oral route, all administrative routes may induce more or less stress on the pigs, leading to deterioration in the production and overall performance of the animal. Vaccine administration routes and their characteristics have been shown in Figure 4. Encouraging results of oral administration of a bread-based lyophilized C-strain of CSFV in backyard pig farming have also been reported [54]. Drinking water vaccines are also increasingly employed in the porcine industry [179]. Low and moderate intranasal and intramuscular vaccine administration can protect the porcine industry against specific viral genomes like ASFV [180]. PEDV-specific IgA and virus-neutralization antibody levels in the colostrum were seen upon oral administration of the live attenuated PEDV (DR13 strain) vaccine [181].

## 9. Adjuvant Systems

Being an integral part of vaccine formulation, the research on adjuvants parallels the vaccine development research. Adjuvants are the immunological or pharmacological substances used along with the vaccine to improve or potentiate the immune response produced due to successful vaccination [182,183]. Based on their characteristics, adjuvants for the veterinary vaccine can be classified into several broad categories, including oil emulsion, particulate antigen carrier, cytokines, pathogen-associated molecular patterns (PAMPs) and immune ligands, liposome nanoparticles, saponins, bacterial cells, toxins, etc. [184,185,186]. One of the significant challenges in porcine vaccine development is finding appropriate adjuvants or combinations of adjuvants [185]. Improved efficacy of the vaccine is noticed upon administration of two or more adjuvants in combination. Saponins and emulsions, when combined, have been shown to strengthen FMD vaccine efficacy [185]. The adjuvants either work as delivery vehicles or as immunostimulants. As a vehicle, adjuvants deliver antigens to the draining lymph nodes and promote antigen uptake by antigen-presenting cells (APCs). Adjuvants working as immunostimulants can activate APCs and direct T-cell differentiation and immunoglobulin isotype switching [184]. Some specific considerations should be followed to improve the immunogenicity and protective efficacy of swine vaccines, such as the stability of immunogen dose, schedule, and mechanism of action (MOA) of the adjuvant(s) used. In addition, the route, adjuvant amount, type of immune response required, safety from adverse reactions, effectiveness, induction period, safety, feasibility to scale up, and cost-effectiveness are some important parameters to consider [184,185].

## 10. Nanotechnology Interventions in Swine Vaccinology

Nanotechnology has revolutionized the vaccine industry due to its ability for site-specific drug delivery and the generation of solid immune protection [187]. Because of their protection from proteolytic degradation, prolonged bioavailability, and maintained slow and sustained antigen release, nanoparticles (NPs) gained importance in vaccine delivery platforms [186,187]. To formulate an effective nano vaccine, the vaccine components can either be encapsulated with NPs or viral antigens are used to decorate the surface of NPs [188]. Compared to soluble antigen vaccines, NP-based vaccines provide a better immune response. Antigen-presenting cells can readily phagocytize NP-loaded antigens, thus imparting different effects like enhanced antigen uptake, augmented antigen processing, induced maturation of dendritic cells, and improved cytokine response [188,189]. In developing nano vaccines, several nanocarriers can be used, including bacterial spores, proteasomes, liposomes, virosomes (liposome and viral envelope protein), superfluids (biodegradable polymer), nanobeads, VLPs, and phages [190]. Nanocarriers generally used in pigs are polymeric nanoparticles, including polysaccharides, polyesters, and chitosan [190,191]. A polyanhydride-nanoparticle-based IAV vaccine was tested in pigs that showed some promising results with high mucosal humoral and cellular immune responses in pigs [192]. Some NP-based vaccine candidates against different pig viruses are listed in Table 4.

NP-based vaccine delivery platforms have great potential, but more research is needed, mainly focusing on NP stability, storage conditions, and immunogenicity. The recent advances in nano vaccines demonstrated in pigs have shown promising results.

## 11. Herd Immunity

When a population becomes immune to a particular disease, it is commonly known as herd immunity. Herd immunity is achieved when enough individuals in a population have developed protective antibodies after recovery from an infection [193]. Before the whole population acquires immunity, a small percentage must contract the disease, called a ‘threshold proportion’, which gives rise to a new term, ‘herd immunity threshold’ or ‘HIT’. The herd immunity threshold is the point at which the population immune to the disease becomes more significant than the threshold proportion, wherein the spread of the disease will decline [194]. HIT can be calculated as follows:HIT = 1 − 1/R0

R0, or the basic reproduction number/rate, estimates the contagiousness and transmissibility of an infectious agent. R0 refers to the average number of individuals that can transmit the infection. Typically, R0 < 1 accounts for the disease being controlled and not spreading. If R0 = 1, one individual can infect one another on average. If R0 > 1, one individual can spread the disease to broader populations, thus leading to an epidemic or pandemic [194,195,196]. In pigs, it has been reported that IM administration of either live, inactivated, or whole virus PEDV vaccine effectively maintains or boosts herd immunity in sows positive for PEDV-specific IgA memory B cells [181].

## 12. Economy, Limitations, and Future Prospective of the Vaccine Industry

The global vaccine market is expected to exceed USD 62 billion by 2027. Difficulties such as research and development, law and intellectual property, and supply and demand intricacies reflect how complex and interconnected vaccine economics is. Vaccine production is costly and capital-intensive [197,198]. Though vaccines came into use many years ago, over time, they must undergo many refinements and modifications, including vaccine safety, potency, and tolerability. Despite technological advancement and high-throughput research and development, specific long-standing challenges and limitations are still associated with vaccine manufacturing and development [199]. Complete protection against a virus/pathogen remains a goal in the porcine vaccine industry. To have a perfect/ideal vaccine candidate, two crucial considerations should be clearly defined for its successful results, i.e., the number of doses and vaccine administration routes. The pre-defined target product profile (TPP), thermostability, delivery systems, molecular adjuvants, enhanced mucosal immunity, stress induced after vaccination, cost-effectiveness, etc., are further improvements that must be considered in future vaccine developments in the porcine industry [6,199].

## 13. Conclusions

This review summarized the status of porcine viral vaccines, highlighting past developments and exploring future opportunities and challenges. Although porcine vaccinology has significantly progressed, the final goal is yet to be achieved. Effective vaccines are available for some of the economically significant porcine viral diseases. However, for viruses like ASFV, there is an urgent need for an effective vaccine. The vaccine industry has been revolutionized with modern vaccine development technologies, but the traditional vaccines still need not be overlooked. The porcine vaccine research should always be aimed toward developing affordable and effective vaccines, which should cater to the specific requirements of pig farming.

## Figures and Tables

**Figure 1 viruses-16-00833-f001:**
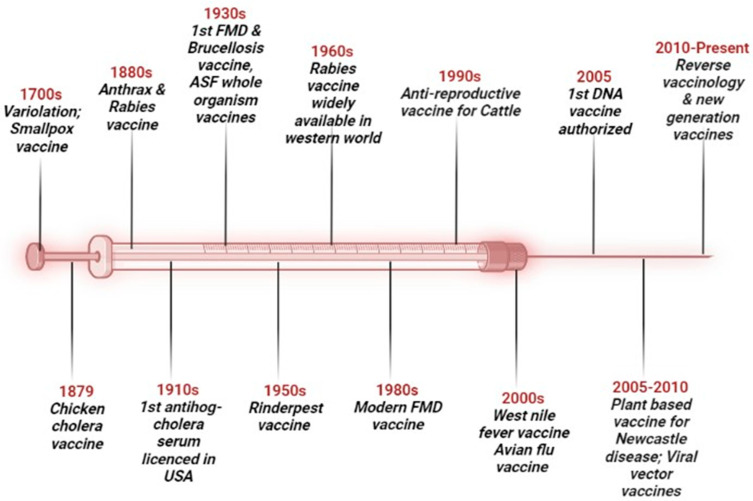
Milestones in animal vaccine development.

**Figure 2 viruses-16-00833-f002:**
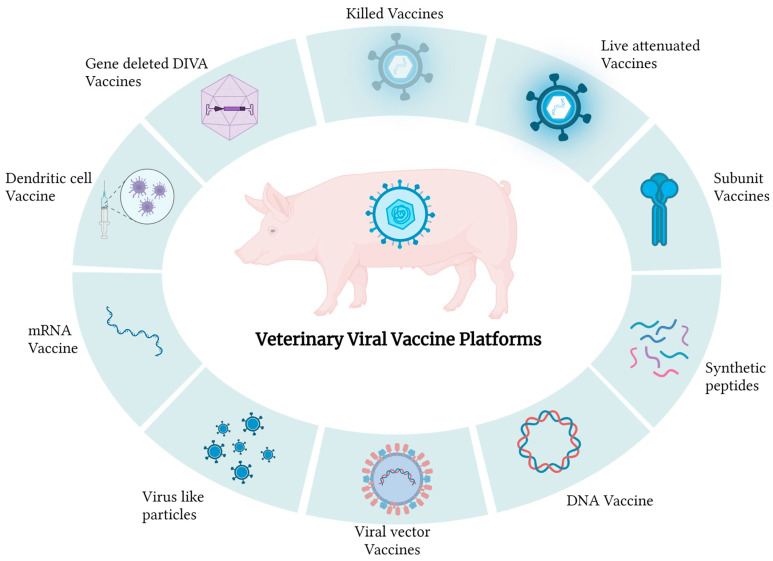
Available veterinary viral vaccines.

**Figure 3 viruses-16-00833-f003:**
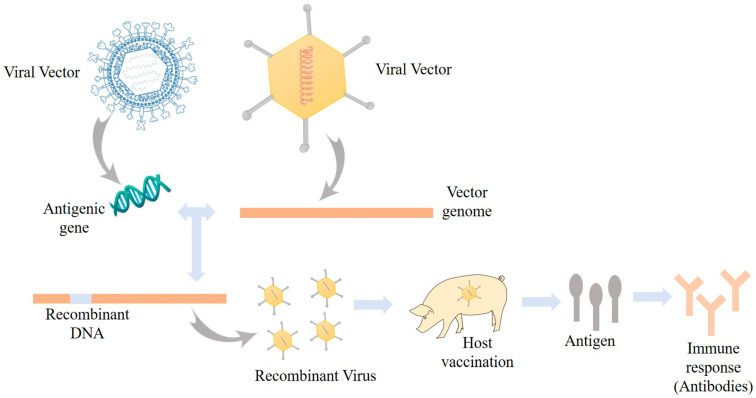
Schematic representation of the working principle of the viral vector vaccine.

**Figure 4 viruses-16-00833-f004:**
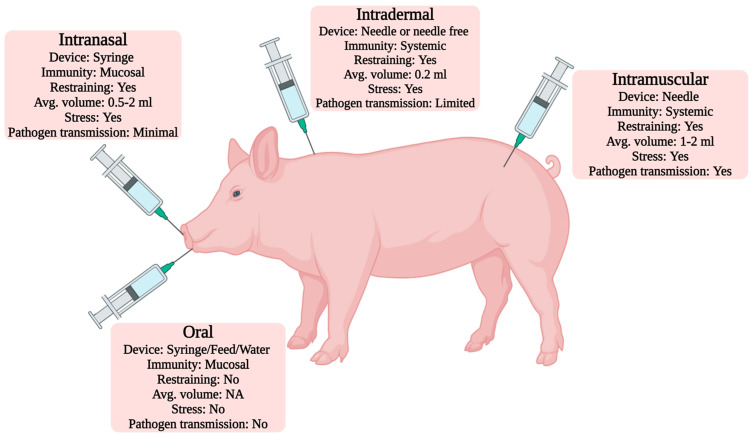
Overview of vaccine administration routes in pigs.

**Table 1 viruses-16-00833-t001:** An overview of the commercially available PRRSV and influenza virus vaccines for pigs, including their manufacturers, types, and administration methods.

Vaccine Name	Manufacturer	Type	Administration Method
Ingelvac^®^ PRRS MLV	Boehringer Ingelheim	Live attenuated	Injectable
Ingelvac^®^ PRRS ATP	Boehringer Ingelheim	Live attenuated	Injectable
ReprosCyc^®^ PRRS EU	Boehringer Ingelheim	Live attenuated	Injectable
Porcilis^®^ PRRS	MSD	Live attenuated	Injectable
Fostera^®^ PRRS	Zoetis	Live attenuated	Injectable
Suvaxyn^®^ PRRS MLV	Zoetis	Live attenuated	Injectable
Prime Pac^®^ PRRS RR	Merck	Live attenuated	Injectable
Prevacent^®^ PRRS	Elanco	Live attenuated	Injectable
UNISTRAIN^®^ PRRS	HIPRA	Live attenuated	Injectable
AMERVAC^®^ PRRS	HIPRA	Live attenuated	Injectable
Ingelvac Provenza™	Boehringer Ingelheim	Live attenuated (H1N1, H3N2)	Intranasal (piglets at one day old)

**Table 2 viruses-16-00833-t002:** Subunit vaccine candidate antigenic proteins for different porcine viruses.

Virus	Subunit Vaccine Candidate Protein
African swine fever virus	p30, p54, p72, pp62, CD2v
Classical swine fever virus	E2
Porcine circovirus (PCV 2)	ORF-2
Foot and mouth disease	VP1, VP2, VP3
Porcine endemic diarrhea	Spike protein (S)
Swine influenza	HA
Porcine reproductive and respiratory syndrome	GP5 and M proteins

**Table 3 viruses-16-00833-t003:** Commercial porcine subunit viral vaccines based on baculovirus expression system.

Virus	Baculovirus-Expression-System-Based Commercial Subunit Vaccines
PCV2	Ingelvac CircoFLEX^®^ (BIVI)
	Porcilis^®^ PCV (MSD)
	Circumvent^®^ PCV (Merck)
PPV	Reprocyc^®^ ParvoFLEX (BIVI)
CSFV	Porcilis Pesti^®^ (MSD Animal Health)
	BayoVac^®^ (BAYER AG)
	Tian Wen Jing (TWJ-E2^®^) (TECON, Shenzhen, China)

**Table 4 viruses-16-00833-t004:** NP-based vaccines against different porcine viruses.

NP Candidate	Porcine Vaccine Targets
VLPs	PRRSV
	IAV
	FMDV
	Encephalomyocarditis virus (EMCV)
	Japanese encephalitis virus (JEV)
	PCV2
	PPV
Poly lactic-co-glycolic acid (PLGA)	PRRSV
	IAV
	PEDV
	IAV
Chitosan	IAV
Nano-11	IAV
	PEDV
Polyanhydride	IAV

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
