# Peer review of "A Comprehensive Review of Our Understanding and Challenges of Viral Vaccines against Swine Pathogens"

_viruses, 2024, doi:10.3390/v16060833_

Round 1

Reviewer 1 Report

Comments and Suggestions for Authors

Overall a comprehensive review on various aspects of  vaccines development and recent applications in swine. The manuscript is informative for the readers suggesting also future development steps for porcine vaccinology. Issues that need alterations are provided in the attached pdf file.

Comments on the Quality of English Language

The English language is fine. Few typos and syntax issues are provided in the attached pdf file.

Author Response

Reviewer comment: Overall, a comprehensive review on various aspects of vaccines development and recent applications in swine. The manuscript is informative for the readers suggesting also future development steps for porcine vaccinology. Issues that need alterations are provided in the attached PDF file.

Author response:

Thank you very much for your valuable suggestions. All the suggestions have been incorporated into our revised submission.

Moreover, the concern raised in the PDF file has also been incorporated. Some of the changes include.

Line 516  “Low and moderate intranasal and intramuscular vaccine administration can protect the porcine industry against specific viral genomes like ASFV”.

Here Low and moderate refers to 103 and 104 TCID50.

Reference: (Sánchez-Cordón PJ, Chapman D, Jabbar T, Reis AL, Goatley L, Netherton CL, Taylor G, Montoya M, Dixon L. Different routes and doses influence protection in pigs immunised with the naturally attenuated African swine fever virus isolate OURT88/3. Antiviral Res. 2017 Feb;138:1-8. doi: 10.1016/j.antiviral.2016.11.021. Epub 2016 Nov 28. PMID: 27908827; PMCID: PMC5245086).

Reviewer 2 Report

Comments and Suggestions for Authors

Dear P.T. Authors,

The manuscript is very well prepared and informative, with good structure and vast references. The colourful figures are attractive and help quickly understand presented issues.

There is only one suggestion I have: the information presented in lines 177-184 about examples of PRRSV live attenuated commercial vaccines could be presented in a table.

My other remarks concern text-editing and are listed in the “Comments on the Quality of English Language”.

Comments on the Quality of English Language

Overall, the manuscript is very well written and the only mistakes detected are editorial.

More specifically:

1.       Abstract, lines 21-22: The 2 sentences are incomplete, please correct.

2.       line 452: Please correct typo mistake: “PDEV”

3.       References: in some references year of publication is not written with bold font, please correct.

Author Response

Reviewer comment: The manuscript is very well prepared and informative, with good structure and vast references. The colourful figures are attractive and help quickly understand presented issues.

There is only one suggestion I have: the information presented in lines 177-184 about examples of PRRSV live attenuated commercial vaccines could be presented in a table.

Author response: The table provides an overview of the commercially available PRRSV and influenza virus vaccines for pigs, including their manufacturers, types, and administration methods. The table has been successfully added in the revised manuscript.

Reviewer comment:  My other remarks concern text-editing and are listed in the “Comments on the Quality of English Language”. Abstract, lines 21-22: The 2 sentences are incomplete, please correct.

Author response: Thank for your valuable comments and the same has been incorporated into the revised manuscript.

Reviewer comment: line 452: Please correct typo mistake: “PDEV”

Author response: The correction has been incorporated in the revision.

Reviewer comment: References: in some references year of publication is not written with bold font, please correct.

Author response: We have rephrased all the references as per the suggestions.

Reviewer 3 Report

Comments and Suggestions for Authors

"It is a well-established fact 17 that vaccination is undoubtedly the most effective strategy to control viral infections in animals". 

I do not think that this statement can be apply for all of infection disease of animals. And talking about "Swine Pathogens" as it mentioned in title it is important to identify the role of vaccination in disease control programs for each pathogen listed in the review. For what reason we need to use vaccine? What are the expectations from vaccination? Can we control disease without vaccination? Also, domestic pigs affected by listed diseases often keep in developing countries. What are the cost of vaccination campaign and infrastructure, requirements to vaccine use, for example cold chain, active surveillance program? Does it really feasible for this countries? What are the examples of successful use of "new generation vaccines" against other animal diseases in the World? It is important to answer this questions before making review on vaccine technologies by itself.

"A comprehensive analysis of publication trends 55 from 1996 to 2016 highlighted ASF, Classical Swine Fever (CSF), Foot and Mouth Disease 56 (FMD), Porcine Circovirus infection (PCV), Porcine reproductive and respiratory syn- 57 drome (PRRS), Pseudorabies, Swine Influenza, and Transmissible Gastroenteritis (TGE) 58 as crucial porcine viral diseases [4]."

From one hand authors pointed of importance of vaccine development and use, but from another hand they do not explain why these diseases are still there? Except African Swine Fever vaccines against other disease available for many years.

"Vaccination is considered the most effective and economical way to prevent viral in- 60 fections." 

I do not agree that vaccines always prevent infection of animals after vaccination. Authors need to explain what do they mean.

"The whole organism vaccines have been successfully tried and used 105 against most of the economically significant swine viral diseases like ASF, swine influ- 106 enza, PRRS, Pseudorabies, Porcine endemic diarrhea, PCV infection, etc. [17]."

Not clear about which successfully use of ASF vaccines authors talking.

In Chapter 3.1. Conventional Killed/ Inactivated Vaccines not clear how correlate the title and the following text:

"African Swine Fever (ASF) has expanded very fast in major pig-producing countries 133 like China, the EU, and Russia after 2007. Its effective control through vaccination might 134 be possible due to the ability of pigs surviving ASF infection to develop protective im- 135 munity against new viral infections [24, 25]. Diverse approaches to developing ASF vac- 136 cines have been tried, however, virion complexity and inability to produce neutralizing 137 antibodies hampered its progress [25]. The classical ASF viral vaccines have also been 138 proven ineffective in inducing specific cytotoxic CD8+ T-cells, which is necessary to elim- 139 inate virus-infected cells. This suggests the design-based approach to address limitations 140 associated with WIV vaccines [24-26]"

"Safety concerns, 175 however, are not expected. The outbreak of PRRS in Danish Pigs in 2020 due to the recom-" 

Do authors really mean that there are no safety concerns for these type of vaccines?

"Live attenuated vaccines (LAVs) are also widely used to control ASF and CSF."

It will be useful to provide the examples of wild use of ASF LAV vaccines for disease control.

". Tremendous improvements have occurred in for- 215 mulating oral vaccines including the baits by absorption of C-strain onto bread followed 216 by subsequent lyophilization [54]. It is stable for 18 months at 4°C, contrary to the com- 217 mercial oral vaccine containing liquid C-strain, which requires -20°C for storage."

But before authors written 

"The live vaccine must be stored at -80⁰C or in lyophilized form and cannot 156 be stored once opened [13, 28]."

Please be consistent.

"However, for viruses like ASFV, there is an urgent need for an effective vaccine. "

But before it was

"Typically, R0 < 1 accounts for the disease being controlled and not 580 spreading".

What is "the basic reproduction number/rate, estimates the contagiousness and transmissibility of" ASFV?

If R0 < 1 for ASFV, why we so urgently need the vaccine?

This manuscript, in my understandin, need to be improve and be more logical.

Comments on the Quality of English Language

English is fine.

Author Response

Reviewer comments: "It is a well-established fact 17 that vaccination is undoubtedly the most effective strategy to control viral infections in animals". 

 I do not think that this statement can be apply for all of infection disease of animals. And talking about "Swine Pathogens" as it mentioned in title it is important to identify the role of vaccination in disease control programs for each pathogen listed in the review. For what reason we need to use vaccine? What are the expectations from vaccination? Can we control disease without vaccination? Also, domestic pigs affected by listed diseases often keep in developing countries. What are the cost of vaccination campaign and infrastructure, requirements to vaccine use, for example cold chain, active surveillance program? Does it really feasible for this countries? What are the examples of successful use of "new generation vaccines" against other animal diseases in the World? It is important to answer this questions before making review on vaccine technologies by itself.

Author response: The concern raised by the reviewer is evident and relevant. The statement regarding vaccination highlighting it as the most effective tool for controlling viral infections is based on the established fact that “prevention is better than cure”. Most of the standard textbooks and literature also highlight the fact that vaccination is an effective tool for controlling the viral infections so much of the work going on in virology is also in the area of developing effective vaccines. However, it is also true that for all viral infections, it is not the vaccine that always helps. There are diseases in animals where still testing and slaughter is the only practical way of control. Also, the economics of vaccination for developing countries is a relevant concern that will be addressed in section 12 of the manuscript. As the review is focused on the swine viral diseases so use of new-generation vaccines in other animal diseases was not touched.

Ref: Page no – 255, Veterinary Immunology: An Introduction, 8th Edition, Ian R. Tizard, 8, CBS Publishers & Distributors, 2009, 8131218163, 9788131218167, 574 pages 

Ref: Page no – 475, Kindt, Thomas J., Richard A. Goldsby, Barbara Anne Osborne, and Janis Kuby. 2007. Kuby Immunology. 6th ed. New York: W.H. Freeman.

Reviewer comments: "A comprehensive analysis of publication trends 55 from 1996 to 2016 highlighted ASF, Classical Swine Fever (CSF), Foot and Mouth Disease 56 (FMD), Porcine Circovirus infection (PCV), Porcine reproductive and respiratory syn- 57 drome (PRRS), Pseudorabies, Swine Influenza, and Transmissible Gastroenteritis (TGE) 58 as crucial porcine viral diseases [4]."

 From one hand authors pointed of importance of vaccine development and use, but from another hand they do not explain why these diseases are still there? Except African Swine Fever vaccines against other disease available for many years.

Author response: In our review, we emphasized the critical role of vaccines in combating various viral diseases within the swine industry. However, it's essential to acknowledge that vaccine administration alone isn't always sufficient to prevent disease outbreaks. A multitude of factors, including stress, fecal microbiota, host genetics, maternal antibodies, exposure to immunosuppressive pathogens, and even external influences like antibiotics and mycotoxins, significantly impact the efficacy of immunization in animals. Despite the availability of numerous vaccines for different diseases, the persistence of outbreaks suggests potential gaps in ensuring proper measures during production, transportation, and administration stages. This discrepancy underscores a crucial point: without addressing these factors comprehensively, the effectiveness of vaccines may be compromised, rendering their use less impactful than intended.

Ref: Augustyniak A, Pomorska-Mól M. Vaccination Failures in Pigs-The Impact of Chosen Factors on the Immunisation Efficacy. Vaccines (Basel). 2023 Jan 19;11(2):230. doi: 10.3390/vaccines11020230. PMID: 36851108; PMCID: PMC9964700.

Reviewer comments: "Vaccination is considered the most effective and economical way to prevent viral in- 60 fections." 

I do not agree that vaccines always prevent infection of animals after vaccination. Authors need to explain what do they mean.

Author response: While we acknowledge the reviewer's observation that vaccines may not always guarantee complete protection against infection in animals, it's important to underscore the undeniable importance of vaccines in combating and managing infectious diseases. Indeed, vaccines have long been recognized as the cornerstone of disease prevention strategies, offering significant benefits in reducing the incidence and severity of illnesses.

However, it's crucial to recognize that the effectiveness of vaccines can be influenced by a range of factors beyond the vaccine itself. As mentioned in the previous comment, considerations such as stress, host genetics, exposure to immunosuppressive pathogens, and other environmental factors all play pivotal roles in shaping the efficacy of a vaccine candidate.

 These factors can impact various stages of the vaccination process, from antigen presentation and immune response to establishing long-term immunity. Therefore, while vaccines remain invaluable tools in disease prevention, it's essential to adopt a holistic approach that addresses not only vaccine development and administration but also considers the broader context in which vaccination occurs. By understanding and mitigating these additional factors, we can optimize the effectiveness of vaccines and enhance overall disease management efforts. 

Ref: Tripathi, T. Advances in vaccines: revolutionizing disease prevention. Sci Rep 13, 11748 (2023). https://doi.org/10.1038/s41598-023-38798-z

Reviewer comments: "The whole organism vaccines have been successfully tried and used 105 against most of the economically significant swine viral diseases like ASF, swine influ- 106 enza, PRRS, Pseudorabies, Porcine endemic diarrhea, PCV infection, etc. [17]."

 Not clear about which successfully use of ASF vaccines authors talking.

 Author response: Thank you for your valuable input here. The necessary changes have been made by removing ASF from the manuscript, as no commercial vaccine is available.

Reviewer comments: In Chapter 3.1. Conventional Killed/ Inactivated Vaccines not clear how correlate the title and the following text:

 "African Swine Fever (ASF) has expanded very fast in major pig-producing countries 133 like China, the EU, and Russia after 2007. Its effective control through vaccination might 134 be possible due to the ability of pigs surviving ASF infection to develop protective im- 135 munity against new viral infections [24, 25]. Diverse approaches to developing ASF vac- 136 cines have been tried, however, virion complexity and inability to produce neutralizing 137 antibodies hampered its progress [25]. The classical ASF viral vaccines have also been 138 proven ineffective in inducing specific cytotoxic CD8+ T-cells, which is necessary to elim- 139 inate virus-infected cells. This suggests the design-based approach to address limitations 140 associated with WIV vaccines [24-26]"

Author response:Thank you for your valuable input here. The necessary changes have been incorporated into the revised manuscript.

Reviewer comments: "Safety concerns, 175 however, are not expected. The outbreak of PRRS in Danish Pigs in 2020 due to the recom-" 

Do authors really mean that there are no safety concerns for these type of vaccines?

Author response: The reviewer has raised the relevant concern. The author wants to communicate that the safety concerns are not common for such type of vaccine, so the ‘expected’ word will be replaced by ‘common’ word in the manuscript to make it more clear.

Reviewer comments: "Live attenuated vaccines (LAVs) are also widely used to control ASF and CSF."

 It will be useful to provide the examples of wild use of ASF LAV vaccines for disease control.

 Author response: Given the concerns raised by the reviewer regarding the use of LAV for ASF, we made a deliberate decision to omit ASF from our discussion. However, it's important to recognize the significance of ongoing research and development efforts aimed at addressing this devastating disease. The emergence of promising candidates like ASFV-G-∆MGF offers hope for the eventual development of a vaccine capable of effectively combating ASF and mitigating its impact on the swine industry.

Moving forward, continued investment in research, rigorous testing, and careful consideration of safety and efficacy will be essential in realizing the potential of these vaccines and ultimately controlling ASF outbreaks. While challenges remain, the progress made thus far underscores the importance of ongoing collaboration and innovation in the field of veterinary medicine.

Ref: Deutschmann, P., Forth, JH., Sehl-Ewert, J. et al. Assessment of African swine fever vaccine candidate ASFV-G-∆MGF in a reversion to virulence study. npj Vaccines 8, 78 (2023). https://doi.org/10.1038/s41541-023-00669-z

Reviewer comments: "Tremendous improvements have occurred in for- 215 mulating oral vaccines including the baits by absorption of C-strain onto bread followed 216 by subsequent lyophilization [54]. It is stable for 18 months at 4°C, contrary to the com- 217 mercial oral vaccine containing liquid C-strain, which requires -20°C for storage."

 But before authors written 

 "The live vaccine must be stored at -80⁰C or in lyophilized form and cannot 156 be stored once opened [13, 28]."

 Please be consistent.

Author response: The live-attenuated vaccines which are not orally administered and require administration through injections need to be stored at lower temperatures to prevent reversion to virulence state, whereas the oral live-attenuated vaccines that are being developed can easily be stored at much high temperatures. Moreover, bread-based lyophilization improves the longevity of vaccines and is tested to be stable at 4°C for up to seven months.

Ref: Opriessnig, T.; Mattei, A.; Karuppannan, A.; Halbur, P., Future perspectives on swine viral vaccines: where are we headed? Porcine Health Management 2021, 7, (1), 1

Kunu, W.; Jiwakanon, J.; Porntrakulpipat, S., A bread‐based lyophilized C‐strain CSF virus vaccine as an oral vaccine in pigs. Transbound Emerg Dis 2019, 66, (4), 1597-1601

Reviewer comments: "However, for viruses like ASFV, there is an urgent need for an effective vaccine. "

 But before it was

"Typically, R0 < 1 accounts for the disease being controlled and not 580 spreading".

Author response: There have been many failed attempts to develop an effective vaccine against ASFV and hence there is an urgent requirement of potent vaccines. Also, R0 < 1 has been observed for viruses like PEDV as mentioned in the manuscript. We have not addressed R0 value for ASFV.

Ref: Jung, K.; Saif, L.; Wang, Q., Porcine epidemic diarrhea virus (PEDV): An update on etiology, transmission, pathogenesis, and prevention and control. Virus research 2020, 286, 198045.

Reviewer comments: What is "the basic reproduction number/rate, estimates the contagiousness and transmissibility of" ASFV?

If R0 < 1 for ASFV, why we so urgently need the vaccine?

Author response: There have been cases of reemergence of the ASFV from time to time which renders it important for us to keep updating our vaccine needs. According to a recent report on global trends of ASFV reproduction number, the R0 value is far beyond the controlled limits (R0=1) and is spreading swiftly in all parts of the world. The highest R0 value was reported in Malta (R0=18) followed by China and Russia (R011). However, in few parts of the world, especially Netherlands the disease is in eradication zone (https://doi.org/10.1155/2024/1046866). It must be noted that the virulence of ASFV depends on the strain and morbidity ultimately depends on the viral load, the greater the viral load the more rapidly it spreads among animals. Yet, the far spread of disease is ultimately a result of human activities. (10.3390/v11090866).

Ref: Sánchez-Cordón, P.; Montoya, M.; Reis, A.; Dixon, L., African swine fever: A re-emerging viral disease threatening the global pig industry. Vet J 2018, 233, 41-48.

Shraddha Tiwari, Thakur Dhakal, Tae-Su Kim, Seong-Hyeon Kim, Sang-Joon Lee, Dae-Sung Yoo, Ho-Seong Cho, Gab-Sue Jang, Yeonsu Oh, "Global Basic Reproduction Number of African Swine Fever in Wild Boar and a Mental Model to Explore the Disease Dynamics", Transboundary and Emerging Diseases, vol. 2024, Article ID 1046866, 7 pages, 2024. https://doi.org/10.1155/2024/1046866

Schulz, K., Conraths, F. J., Blome, S., Staubach, C., & Sauter-Louis, C. (2019). African Swine Fever: Fast and Furious or Slow and Steady?. Viruses, 11(9), 866. https://doi.org/10.3390/v11090866

Reviewer 4 Report

Comments and Suggestions for Authors

Line 107 and table 1=  Porcine endemic diarrhea  should be write Porcine epidemic diarrhea

Line  129-130 explain more in detail the " -vaccine associated respiratory disease in vaccinated pigs (23)"

 Line  221  Live attenuated DIVA vaccines added  DIVA (Differentiated infected-vaccinated antibodies)

Line  389  MHC  added  (Major Histocompability Complex)

Line  396  CMI added (Cell Mediated Immmune Response)

Line  409.  5.2 mRNA vaccines ( according the the extensive information related with this vaccine against SARS-CoV 2  I think  it should be expanded a little more.

Line  470 VPL  clarify what it means  

Line  762 reference 73  the reference journal in italic 

Author Response

Reviewer comment: Line 107 and table 1= Porcine endemic diarrhea should be written Porcine epidemic diarrhea

Author response: We agree with the reviewer suggestion. The same has been incorporated in the revised version.

Reviewer comment: Line  129-130 explain more in detail the " -vaccine associated respiratory disease in vaccinated pigs (23)"

Author response:  The suggestion has been incorporated in the revised manuscript with additional reference.

Vaccine-associated enhanced diseases (VAED) are altered forms of clinical infections observed in individuals who have previously been vaccinated against a specific pathogen and subsequently exposed to the wild-type version of that pathogen. Classic instances of VAED include atypical measles and enhanced respiratory syncytial virus (RSV) infections following administration of inactivated vaccines for these diseases.

Reviewer comment:  Line  221  Live attenuated DIVA vaccines added  DIVA (Differentiated infected-vaccinated antibodies)

Author response: The DIVA full form is written in the line no 201 of the revised manuscript.

Reviewer comment: Line  389  MHC  added  (Major Histocompability Complex)

      Author response: The changes has been incorporated in the revised manuscript.

Reviewer comment: Line  396  CMI added (Cell Mediated Immmune Response)

Author response: The changes has been incorporated in the revised manuscript.

Reviewer comment: Line  409.  5.2 mRNA vaccines ( according the the extensive information related with this vaccine against SARS-CoV 2  I think  it should be expanded a little more.

Author response: Thank you very much for the valuable comments. We have added the  detailed information about mRNA vaccines in the revised manuscript with additional references.

Reviewer comment: Line  470 VPL  clarify what it means  

Author response: VLPs stands for virus like particles. We have written it in Line 419 please check.

Reviewer comment: Line  762 reference 73  the reference journal in italic 

Author response: The changes has been incorporated in the revised manuscript.

Round 2

Reviewer 3 Report

Comments and Suggestions for Authors

In general I am satisfied with authors comments.

But the statement that commercial vaccines against ASF are not available is out of date.

Author Response

Reviewer Comment: But the statement that commercial vaccines against ASF are not available is out of date.

Author response: Thank you for your valuable comment. The changes have been made in the revised manuscript.
